# Interannual and Seasonal Vegetation Changes and Influencing Factors in the Extra-High Mountainous Areas of Southern Tibet

**Zu-Xin Ye** [1,2,3], **Wei-Ming Cheng** [2] , **Zhi-Qi Zhao** [1,4,]*, **Jian-Yang Guo** [1], **Hu Ding** [1] and **Nan Wang** [2,3]

[1] State Key Laboratory of Environmental Geochemistry, Institute of Geochemistry, Chinese Academy of Sciences, Guiyang 550081, China; yezuxin@mail.gyig.ac.cn (Z.-X.Y.); guojianyang@vip.skleg.cn (J.-Y.G.); Dinghu@vip.skleg.cn (H.D.)

[2] State Key Laboratory of Resources and Environmental Information System, Institute of Geographical Sciences and Natural Resources Research, Chinese Academy of Sciences, Beijing 100101, China; chengwm@lreis.ac.cn (W.-M.C.); wnan@lreis.ac.cn (N.W.)

[3] University of Chinese Academy of Sciences, Beijing 100049, China

[4] School of Earth Science and Resources, Chang'an Univeristy, Xi'an 710054, China

\* Correspondence: zhaozhiqi@chd.edu.cn

**Abstract:** The ecosystem of extra-high mountain areas is very fragile. Understanding local vegetation changes is crucial for projecting ecosystem dynamics. In this paper, we make a case for Himalayan mountain areas to explore vegetation dynamics and their influencing factors. Firstly, the interannual trends of the normalized difference vegetation index (NDVI) were extracted by the Ensemble Empirical Mode Decomposition (EEMD) algorithm and linear regression method. Moreover, the influence of environmental factors on interannual NDVI trends was assessed using the Random Forests algorithm and partial dependence plots. Subsequently, the time-lag effects of seasonal NDVI on different climatic factors were discussed and the effects of these factors on seasonal NDVI changes were determined by partial correlation analysis. The results show that (1) an overall weak upward trend was observed in NDVI variations from 1982 to 2015, and 1989 is considered to be the breakpoint of the NDVI time series; (2) interannual temperature trends and the shortest distance to large lakes were the most important factors in explaining interannual NDVI trends. Temperature trends were positively correlated with NDVI trends. The relationship between the shortest distance to large lakes and the NDVI trend is an inverted U-shaped; (3) the time-lags of NDVI responses to four climatic factors were shorter in Autumn than that in Summer. The NDVI responds quickly to precipitation and downward long-wave radiation; (4) downward long-wave radiation was the main climate factor that influenced NDVI changes in Autumn and the growing season because of the warming effect at night. This study is important to improve the understanding of vegetation changes in mountainous regions.

**Keywords:** NDVI; interannual; seasonal; EEMD; random forests; Time-lag effect

## 1. Introduction

As an intermediate link between hydrosphere, atmosphere, and lithosphere, vegetation plays a critical regulatory role in carbon cycling and reducing greenhouse gas emissions [1,2]. In the context of the warming effect, the response of vegetation to climate change has received much more attention [3–5]. Understanding the variations of vegetation activity, and its driving factors, is of great interest to the assessment of regional environmental conditions.

Over the past three decades, most parts of vegetated land showed a greening trend, especially in the middle and high latitudes of the North Hemisphere, while browning trends were observed in South Hemisphere [6–8]. At the regional scale, Sahara, U.S. Great plain, and some other regions have been found to have a greening trend [9,10], whereas browning trends were found in the Northeast China Plain, Madagascar savanna, and Amazon region [11–13]. Temperature and precipitation have been considered critical driving factors in many studies on vegetation change [6]. Park and his co-authors [11] reported that precipitation and temperature affect vegetation growth and degradation in East Asia. However, other factors that have not been extensively explored include downward radiations and topographic factors. Downward radiation is the primary energy source of vegetation, and topographic factors determine the hydrothermal conditions of soil to some extent [12]. The impacts of downward long-wave radiation, downward short-wave radiation, and topographical factors on vegetation changes are also crucial in mountainous ecosystems. By introducing these factors, the research will deepen our understanding of the relationship between interannual normalized difference vegetation index (NDVI) trends and changing environmental conditions [13].

In addition to interannual variations, other time scales should be considered, such as seasons, because of seasonal climate changes. In the Tibet Plateau, annual precipitation is centralized in summer due to the Indian monsoon, whereas the precipitation decreased in autumn and winter due to the westerly circulation [14]. For temperature in the Tibet Plateau, the warming rate in autumn is faster than that in summer and spring [15]. The vegetation has different growth rates in spring, summer, and autumn [16]. Therefore, vegetation changes and influencing factors on the seasonal scale must be accounted for. In the background of climate changes, topographic factors have relative stability, while precipitation, temperature, downward radiation, and other meteorological factors are in the dynamic change process. The relationship between climate factors and NDVI variations reveals the driving force of seasonal vegetation changes.

Before analyzing the relationship between seasonal vegetation variations and climate factors, the time-lag effect should not be neglected. The time-lag effect of vegetation responses on climate change is caused by the constraints of various other factors (soil properties, topography, etc.) on vegetation changes. The current vegetation activities may be affected by early climate change. The time-lag effect of vegetation responses to climate change varies with seasons [17] and climate factors [18,19]. Time-lag effects should be considered to improve the accuracy of evaluating the relationships between vegetation activities and climatic factors.

Although there are many studies on vegetation changes, fewer studies were concentrated in the mountain area at mid-high latitudes. The Himalayas extra-high mountain region (HEM) is a typical mountainous areas at mid–high latitudes with a higher warming rate and abundant geomorphological forms, such as valleys, platforms, hills, and mountains [20]. The ecological diversity of the HEM region decreases from east to west, changing from the Yarlung Zangbo River Gorge with high biodiversity in the alpine grassland with less biodiversity [21]. Simultaneously, the southeast portion has experienced forest degradations [22], and the vertical zonal effect of vegetation is evident in the northeast portion [23]. The HEM region is a typical mountainous area. As such, assessing changes in vegetation in this region will improve our overall understanding of vegetation change in mountainous ecosystems.

This paper focuses on the spatiotemporal changes in vegetation activities and their responses to environmental factors in the mountainous areas. The aims of this paper are to (1) assess the interannual trends in vegetation changes, (2) investigate the time-lag effects of vegetation activities responses to climatic factors, and (3) analyze the relationship between vegetation changes and environmental factors. The findings of this paper were used to provide a reference for the ecological security in the HEM region.

## 2. Materials and Methods

### 2.1. Study Area

The Himalayas extra-high mountain region (HEM) is located in southern Tibet (Figure 1a). It stretches approximately 1700 km from west to east and 1000 km from south to north, with an average elevation above 4000 m.

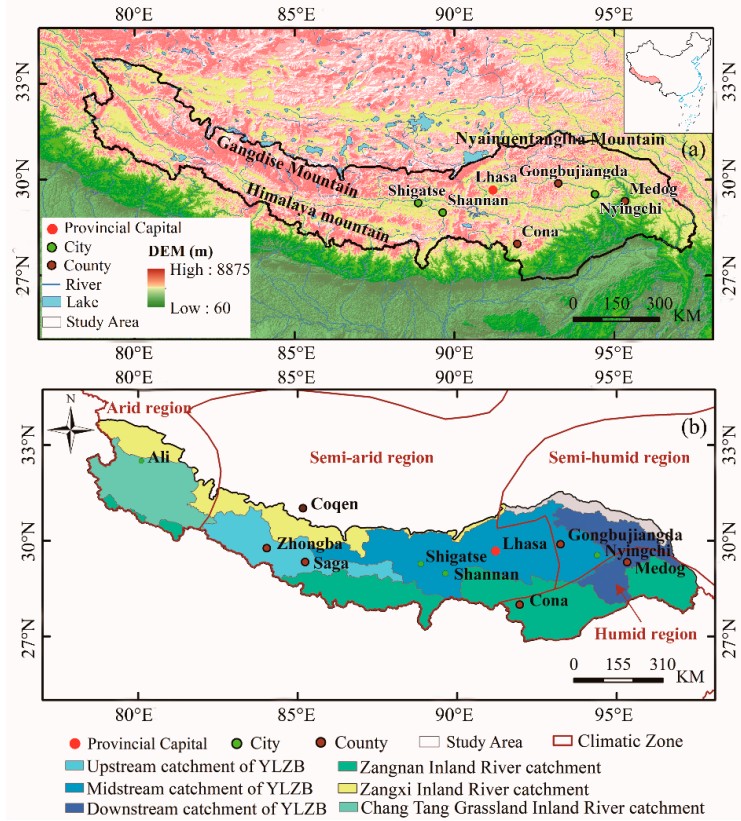

**Figure 1.** (**a**) Location of the Himalayas extra-high mountain region (HEM) region in China; (**b**) geographical distribution of the climate zones and three-level catchments over the HEM region.

From east to west, the HEM region has experienced a substantial climate changes from a humid climate to a semi-humid climate, semi-arid climate, and arid climate (Figure 1b). Medog county and Cona county are situated in the humid climate region with annual precipitation above 500 mm [24] and mainly covered with broad-leaved forests and needle-leaved forests. Gongbujiangda county is located at the transition from southern Tibet valley to eastern Tibet alpine valley, with a mild and humid climate in its eastern portion and a cold and dry climate in its western part. Coqen, Zhongba, and Saga counties belong to semi-arid regions with annual precipitation ranging from 200 to 300 mm and characterized by large day-night temperature differences and long light duration. Alpine vegetation, grasslands, and meadows account for 80% of total areas in the semi-arid region. The HEM region is rich in water resources and contains six catchments, including the upstream, midstream, and downstream catchments of Yarlung Zangbo River, Zangnan inland river catchment, Zangxi inland river catchment, and the Chang Tang grassland inland river catchment.

### 2.2. Data Source

#### 2.2.1. Global Inventory Modelling and Mapping Studies (GIMMS) NDVI

For this study, we utilized the freely available GIMMS3g Normalized Difference Vegetation Index (NDVI) data. The GIMMS3g NDVI with a temporal resolution of 15 days and spatial resolution of

0.0833° was downloaded from the National Aeronautics and Space Administration (http://ecocast.arc.nasa.gov/data/pub/). This data has a longer time series (1982–2015) and a more extensive spatial range compared with that of MODIS NDVI [25] and SPOT VGT NDVI [26]. The errors derived from volcanic eruptions, solar elevation angles, and sensor sensitivity were removed from the GIMMS NDVI data [27]. The artifacts caused by orbital drift and variations in solar angle and view zenith angle were calibrated during post-processing [28], and the cloud effects were reduced by using the highest fortnightly value within 0.0833° of pixels. To match the time and spatial resolution of the meteorological data, GIMMS NDVI was used to generate monthly average NDVI data and resample the data to 0.05° using the nearest neighborhood method. Water and permanent snow were also included in the insignificant group to avoid their impact on the pixels. Meanwhile, pixels with an NDVI value of lower than 0.1 for the growing season (May to October) were thought to be non-vegetation pixels and classified as the insignificant group [29].

### 2.2.2. Meteorological Data

For this study, we utilized the China Meteorological Forcing Data (CMFD), including temperature, precipitation, downward long-wave radiation, and downward short-wave radiation. These data are available from the Cold and Arid Regions Science Data Center (westdc.westgis.ac.cn/data). The CMFD with a time span of 1979~2015 has a three hour and 0.1° temporal–spatial resolution. The data combines five auxiliary data sources: China Meteorological station data, Tropical Rainfall Measuring Mission (TRMM) 3B42 precipitation data, Global Energy and Water cycle Experiment- Surface Radiation Budget project (GEWEX-SRB) downward shortwave radiation data, Princeton forcing data, and Global Land Data Assimilation System (GLDAS) data. The CMFD has been widely utilized in primary plant productivity estimation [30], driving factor analysis of vegetation growth [31], and lake area simulation [32]. Before the utilization, the 3-hour interval meteorological data were aggregated into the monthly average temperature, monthly total precipitation, monthly total downward long-wave radiation, and monthly total downward short-wave radiation data. Meanwhile, the processed data were resampled to 0.05° using the nearest neighborhood method.

### 2.2.3. Geographic Data

Derived from the Shuttle Radar Topography Mission (SRTM) conducted by NASA and NGA, the global digital elevation model (DEM) with a spatial resolution of 90 m was downloaded from the website of the U.S. Geological Survey (USGS) (http://earthexplorer.usgs.gov). The fishnet with a spatial resolution of 0.05° × 0.05° extracted the mean DEM based on the mean function of zonal statistics from the ArcGIS software. The fishnet was converted to a raster with a spatial resolution of 0.05° based on the attribute of mean DEM. The third-level catchment boundary in vector format was downloaded from the website of the Resource and Environment Data Cloud Platform (http://www.resdc.cn/).

### *2.3. Methodology*

### 2.3.1. Ensemble Empirical Mode Decomposition (EEMD)

The EEMD algorithm was used to extract the annual components of NDVI and climatic factors at the whole HEM scale and pixel scale. Taking NDVI as an example, the monthly average NDVI values from 1982 to 2015 were calculated to generate a time series $X(t)$ with 408 values. Then, the time series of $X(t)$ was input into the EEMD algorithm to generate m IMF components and one residual (Equation (1)). Each IMF has its own mean period T, which can be calculated by Equation (1). Based on the grouping criteria proposed by Wen et al. [33], we summed one residual and the IMFs with a mean period T greater than 2 to obtain the interannual variation component. If it is needed to obtain the interannual trend, linear regression was applied for the annual variation component. At a pixel scale, the same method was used to extract the interannual variation component pixel-by-pixel:

$$T_i = 2X_i/n \tag{1}$$

where $T_i$ means the average time period of $IMF_i$, year; $X_i$ means the length of time series of $IMF_i$, year; n is the number of extrema points of $IMF_i$.

EEMD is an adaptive time-frequency data analysis method developed from Empirical Mode Decomposition (EMD) [34]. EMD can extract a finite number of components (named IMFs) from nonstationary and nonlinear data series. The process can be written as Equation (2), in which X(t) means the original data sequence and r(t) is the residual term. Each IMF has its own respective frequency and periodic change character. All IMFs meet the following two conditions: (1) the number of extreme points and zero-crossing points must be equal or different at most one in the whole time series, and (2) the average value of the envelopes corresponding to the local maxima and local minima is zero at any point.

$$X(t) = \sum_{i=1}^{m} IMF_i(t) + r(t). \tag{2}$$

Based on the framework of the EMD method, EEMD introduced an ensemble of white noise signals to overcome the problem caused by mode mixing. Compared with the commonly used methods for time series analysis, such as Fourier spectral analysis and Breaks for Additive Season and Trend (BFAST), the EEMD does not need an a priori-defined basis function. Because of its flexibility and adaptability, the EEMD method has been applied to many studies, such as climatic data analysis [35,36] and ecosystem changes [33,37]. Here, the EEMD algorithm was applied to extract the interannual component of NDVI and climate variables.

### 2.3.2. Breaks for Additive Season and Trend (BFAST)

In this study, the BFAST algorithm was used to detect the breakpoint in time series of the interannual variation component of NDVI. The interannual variation component of NDVI at the whole HEM scale was input in the BFAST algorithm. Because the seasonal component was removed when the EEMD method was used to extract the interannual component, the season model of the BFAST algorithm was set to "NONE". Then, the parameter of h was set to 1/7, according to previous studies [38,39]. The maximum number of breakpoints was set to one, avoiding many breakpoints that could complicate the results. Meanwhile, one-breakpoint test can ensure that only the most significant change in the time series is detected [40]. Finally, the algorithm output the most significant breakpoint in the interannual variation component. BFAST analysis was performed in the R environment and the corresponding package was downloaded from the website (https://cran.rproject.org/web/packages/bfast/index.html). The BFAST algorithm is a statistically-based breakpoint analysis method and has widely been applied in the breakpoint detection of satellite image time series [38,41,42].

### 2.3.3. Random Forest Regression

To evaluate the importance of environmental predictors on interannual NDVI trends, a random forest regression model between NDVI and nine environment factors was used. The nine environment variables are described in Table 1. For 19416 pixels within the HEM region, there was information about interannual NDVI trends and nine environment variables. Because there is no need for the prediction function of the Random Forest, this data (n = 19416) does not need to be randomly split into test data and validation data. The data (n = 19416) were fully inputted into the Random Forest model, and the interannual NDVI trends were used as the response variable. The randomly selected variables at each node (mtry), and the number of regression trees of a bootstrap sample (ntree), were set to 4 and 500, respectively.

The nine environmental factors selected in this paper come from three aspects—climate, topography, and geography. Among the climatic factors, temperature and precipitation have been crucial factors for vegetation changes in previous studies [6,11]. Decreased downward radiation is

considered to be the cause of much forest land degradation, such as the degradation of the southeastern Tibet plateau [22]. Elevation and slope are critical topographic factors. Increasing elevations will significantly accelerate the climate's warming rate [43], and slope changes can affect soil erosion conditions. Lakes and rivers are important water resources for vegetation in arid and semi-arid regions, and the distances to lakes and rivers will affect the vegetation's growth environment. The catchment is the basic hydrological unit, and the hydrological conditions between catchments are quite different, which can lead to the spatial heterogeneity of vegetation changes [44,45].

Random Forest is a combination of tree predictors, such that each tree is built from a bootstrap sample of the original data [46,47]. Random Forest Regression (RFR) can be presented by Equation (3), in which $\theta_t$ means an independent identically distributed random vector, x means an input vector, and T means the number of trees:

$$h(x) = \frac{1}{N} \sum_{t=1}^{T} \{h(x, \theta_t)\}. \tag{3}$$

The Random Forest Regression method was applied in this study to analyze the importance of different environmental factors on interannual NDVI trends. The main evaluation index for the importance is mean decrease accuracy (%IncMSE). The %IncMSE means the percent increase in MSE as a result of the variable being randomly permuted. More important factors have higher values of %IncMSE.

The partial dependence plot enhances the exploratory function of the Random Forest model. This plot provides the ability to visualize the relationships between the response variable and many explanatory variables [48] and can be used directly from the Random Forest package on the R platform. In this paper, we used the partial dependence plot to visualize the relationships between interannual NDVI trends and nine environmental factors.

**Table 1.** Preprocessing of the nine environmental factors for the Random Forest model.

| Factor | Description |
|---|---|
| Interannual temperature trend | Firstly, the interannual variation component of temperature was extracted by the ensemble empirical mode decomposition (EEMD) algorithm. Subsequently, the linear regression was applied for the interannual variation component to obtain the interannual trend of temperature (°C·year$^{-1}$). Each pixel has one interannual trend. |
| Interannual precipitation trend | The interannual precipitation trend (mm·year$^{-1}$) was obtained by the same method as above. |
| Interannual downward long-wave radiation trend | The interannual downward long-wave radiation trend (W·m$^{-2}$·year$^{-1}$) was obtained by the same method as above. |
| Interannual downward short-wave radiation trend | The interannual downward short-wave radiation trend (W·m$^{-2}$·year$^{-1}$) was obtained by the same method as above. |
| Elevation | The digital elevation model (DEM) with a spatial resolution of 0.05° |
| Slope | Slope (°) was calculated from DEM through the surface analysis function of the ArcGIS software |
| Distance to rivers | Euclidean distance (m) to the nearest rivers > 100 m |
| Distance to large lakes | Euclidean distance (m) to the nearest lakes > 1000 m$^2$ |
| Catchment | China's third-level catchment boundary was used in this study. The HEM region was divided into seven sub-regions, namely the Chang Tang Grassland Inland River catchment, the upstream catchment of the Yarlung Zangbo River, the midstream catchment of Yarlung Zangbo River, the downstream catchment of Yarlung Zangbo River, the Zangxi Inland River catchment, and the Zangnan Inland River catchment. |

EEMD: Ensemble Empirical Mode Decomposition.

### 2.3.4. Simple Linear Regression

Simple linear regression was established to determine the correlation between NDVI and each climatic factor, namely, temperature (TEMP), precipitation (PRE), long-wave radiation (LR), and short-wave radiation (SR). The relationships between NDVI and climatic factors are shown in Equations (4)–(7):

$$NDVI = a_i \times TEMP + b \tag{4}$$

$$NDVI = a_i \times PRE + b \tag{5}$$

$$NDVI = a_i \times LR + b \tag{6}$$

$$NDVI = a_i \times SR + b \tag{7}$$

where $a_i$ means the regression coefficient with a time lag of $i$ month. $i$ ranges from 0 to 5 months, while 0 represents no time-lag effect and 1–5 represents 1–5 months lag. Some studies found that the time-lag effects of NDVI responses to climate variables are generally shorter than half a year in Tibet [18,49].

Summer was taken as an example. When the time lag $i = 0,1, \ldots ,5$ months, the correlation coefficients ($R^2$) between NDVI and each climatic factor were calculated. These correlation coefficients ($R^2$) were divided into four levels according to the magnitude of their absolute values: Low (<0.3), Medium (0.3–0.5), High (0.5–0.8) and Very-High (>0.8). For each climatic factor and each time lag, we calculated the area fractions corresponding to four levels.

2.3.5. Partial Correlation Analysis

Partial correlation is defined as the correlation of two factors controlling the influence of the other factors [50]. At the pixel scale, monthly NDVI and monthly climatic factors (temperature, precipitation, LR, SR) were input into the partial correlation analysis model in MATLAB, and then the partial correlation coefficients between NDVI and each climate factor were output. Subsequently, the climatic factor with the largest partial coefficient was selected as the main influence factor. The specific formula for partial correlation is described as follows:

$$r_{xy-z} = \frac{r_{xy} - r_{xz} \times r_{yz}}{\sqrt{1 - r_{xz}^2} \times \sqrt{1 - r_{yz}^2}} \tag{8}$$

where $r_{xy-z}$ is the partial correlation coefficients between x variable and y variable when the z variable is selected as the control factor. The $r_{xy}$, $r_{xz}$ and $r_{yz}$ are the correlation coefficients between x and y, x and z, and y and z, respectively.

**3. Results**

*3.1. Interannual Trends of NDVI and Climate Factors*

Figure 2a illustrated the interannual NDVI variation at the whole HEM scale from 1982 to 2015. The overall NDVI exhibited a positive trend between 1982 and 2015 at a speed of 0.00012 year$^{-1}$ ($p < 0.05$), slower than the growth of Tibet Plateau (0.0002 year$^{-1}$) [51]. The breakpoint in interannual NDVI variations appeared in 1989 (Figure 2b). Before the breakpoint time, NDVI exhibited a consistent positive trend with a rate of 0.0015 year$^{-1}$ ($p < 0.01$). After the breakpoint time, a negative trend with a speed of 0.00044 year$^{-1}$ ($p < 0.01$) was found. At the pixel scale, significant positive trends of interannual NDVI components were found in Gongbujiangda county and surrounding areas and weak positive trends with a speed between 0 and 0.0005 year$^{-1}$ were found in the central and northwestern HEM region (Figure 3). Pixels with a negative trend of interannual NDVI components were concentrated in the southeast HEM region.

Trend patterns of climate factors were spatially heterogeneous (Figure 4). For interannual temperature change, most areas showed a positive trend with a rate between 0 and 0.035 °C per year$^{-1}$. The largest positive trends for temperature were found in Zhongba county, Saga county, Coqen county and Ali Prefecture. For interannual precipitation change, the largest positive trends were found in the southeast HEM region, and the negative trends were found in Lhari county, Saga county, and Coqen county. For downward long-wave radiation and downward short-wave radiation, most pixels within

the HEM region showed a positive trend of interannual long-wave radiation variations and a negative trend of interannual short-wave radiation variations.

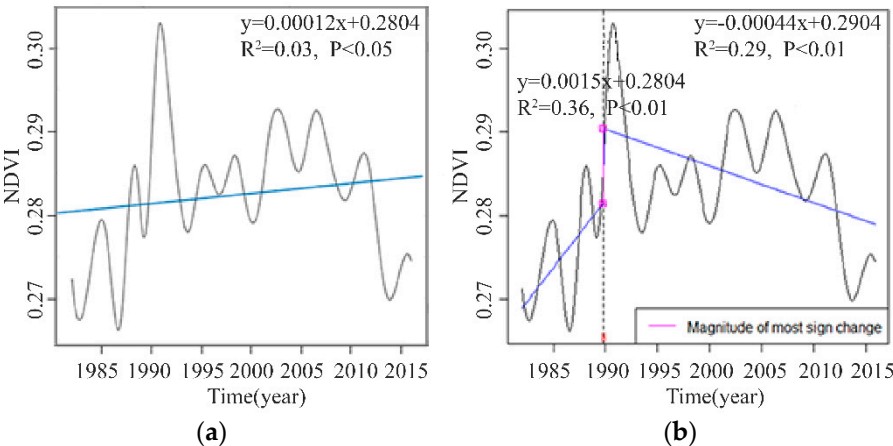

**Figure 2.** Interannual components of normalized difference vegetation index (NDVI) extracted from a spatial average time series at the whole HEM scale. for the period of 1982~2015 (**a**), and before and after the breakpoint time of 1989 (**b**).

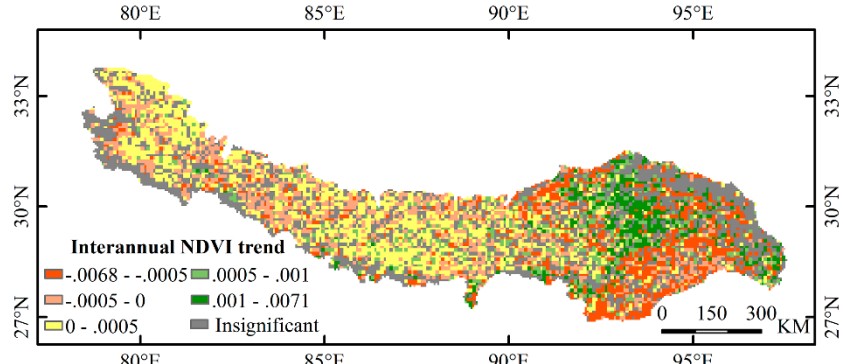

**Figure 3.** Spatial pattern of interannual NDVI trends at a pixel scale from 1982 to 2015.

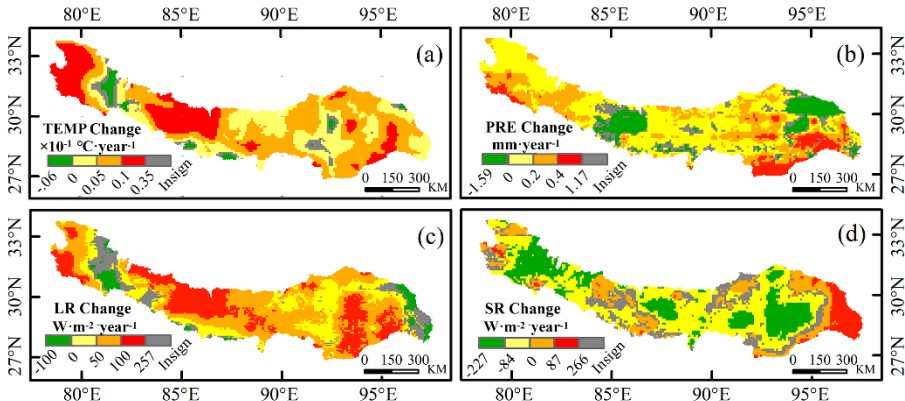

**Figure 4.** Spatial patterns of the interannual trends of four climate factors: temperature (**a**), precipitation (**b**), downward long-wave radiation (**c**), and downward short-wave radiation (**d**).

*3.2. Environment Influences on Interannual NDVI Trend*

Figure 5 shows the importance magnitude of nine environmental variables. Temperature and the shortest distance to large lakes were the dominant factors in determining the direction and magnitude of interannual NDVI trends. Downward long-wave radiation, downward short-wave radiation, elevation,

slope, and the shortest distance to rivers had moderate values of %IncMSE, indicating they are also important in influencing the NDVI trends. There was a decline in importance for precipitation and river catchments.

We found different relationships between environmental factors and interannual NDVI trends (Figure 6). The increased warming leads to the large positive trend of NDVI. When the warming rate is higher than 0.013 °C·year$^{-1}$, the NDVI trend remains unchanged. The increased precipitation trend leads to a larger positive trend of NDVI. Elevation has been considered to have a strong association with the NDVI trend. The large positive trend of NDVI is found in high altitude areas. However, when elevation exceeds 5000 m, the NDVI trend starts to decrease. For downward long-wave radiation (LR) and downward short-wave radiation (SR), the long-wave radiation trend has an apparently negative relationship with the NDVI trend and short-wave radiation has an uncertain relationship with the NDVI trend. The shortest distance to large lakes plays a vital role in determining the magnitude and direction of the interannual NDVI trend. The curve between the shortest distance to large lakes and the NDVI trend presents an inverted U-shape. Within 20 km, the NDVI trend increases significantly, with an increase in the shortest distance to large lakes. When the shortest distance to large lakes exceeds 20 km, the NDVI trend decreases significantly, with an increase in the distance to large lakes. Among six river catchments, the midstream catchment of the Yarlung Zangbo river exhibits the largest positive NDVI trend and the downstream catchment of the Yarlung Zangbo river exhibits the most significant negative NDVI trend.

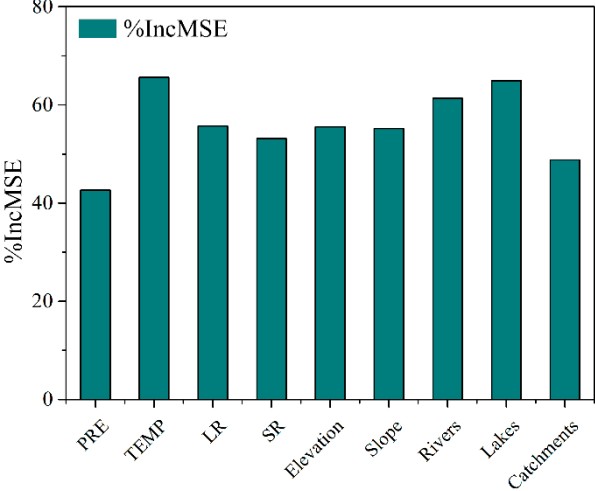

**Figure 5.** The importance of environment factors from the Random Forests model. The eight variables on the x-axis are precipitation (PRE), temperature (TEMP), downward long-wave radiation (LR), downward short-wave radiation (SR), elevation, slope, shortest distance to rivers, shortest distance to large lakes and river catchments. Mean Decrease Accuracy (%IncMSE) is the variable on the y-axis.

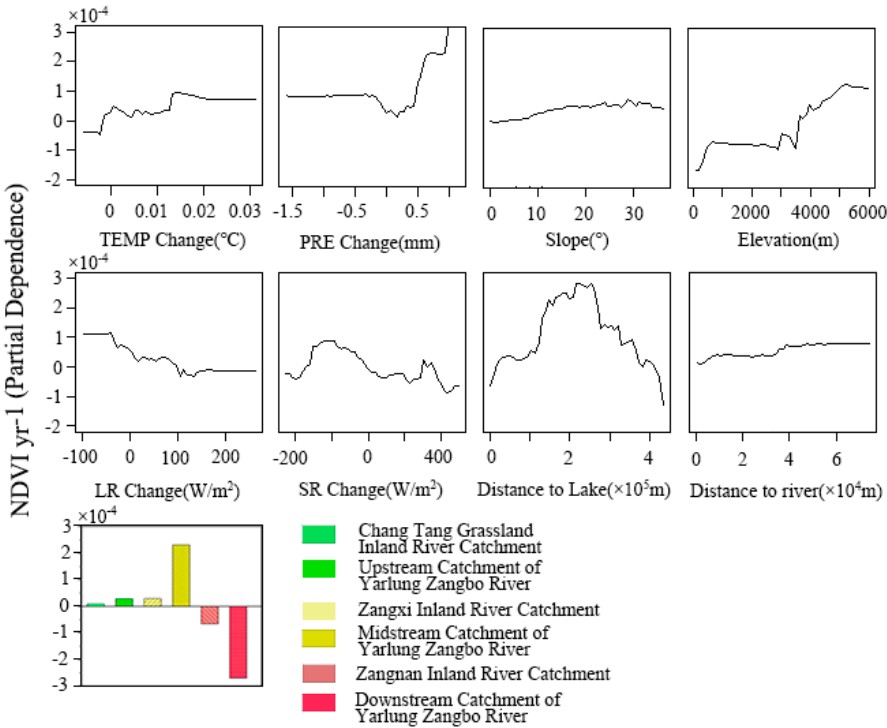

**Figure 6.** Partial dependence plots of nine environment factors for the whole HEM region. These plots represent the influence of environmental factors (x-axis) on the interannual NDVI trend (y-axis).

### 3.3. Relationships between Climate Variables and NDVI for Different Seasons

### 3.3.1. Time-Lag Effects of Vegetation Responses to Climatic Factors at a Seasonal Scale

The time-lag effects of vegetation responses to four climatic factors, that is, temperature (TEMP), precipitation (PRE), downward long-wave radiation (LR), and downward short-wave radiation (SR), were obtained using the NDVI time-series and China Meteorological Forcing Data. The results showed that the time-lag effects varied with climatic factors and seasons. For Summer, when the time lag of NDVI responses to temperature is two months, the area fraction corresponding to the high level of the correlation coefficient reaches its largest size, which indicates that the optimal time lag of NDVI responses to temperature is two months (Figure 7a). In Autumn, the optimal time lag of NDVI responses to temperature (0 month) is shorter than that in Summer (two months), possibly because the temperature in Autumn drops rapidly, and the demand for the ideal temperature suitable for vegetation growth is increasing. The optimal time lags of NDVI to temperature in Summer and the growing season are same, which indirectly indicates that vegetation activities in summer are crucial for the growing season. The NDVI exhibited a certain time-lag effect related to precipitation in different seasons, with the time lag in most areas equaling one month. LR and SR are the primary forms of energy input on the Earth's surface. Figure 7g–i shows that the optimal time lags of NDVI responses to LR are one month, 0 month, and one month, in Summer, Autumn, and the growing season, respectively. Figure 7j–l show that the time-lag effects of NDVI respond to SR. The optimal time lags in summer, autumn, and the growing season are two months, 0 months, and four months.

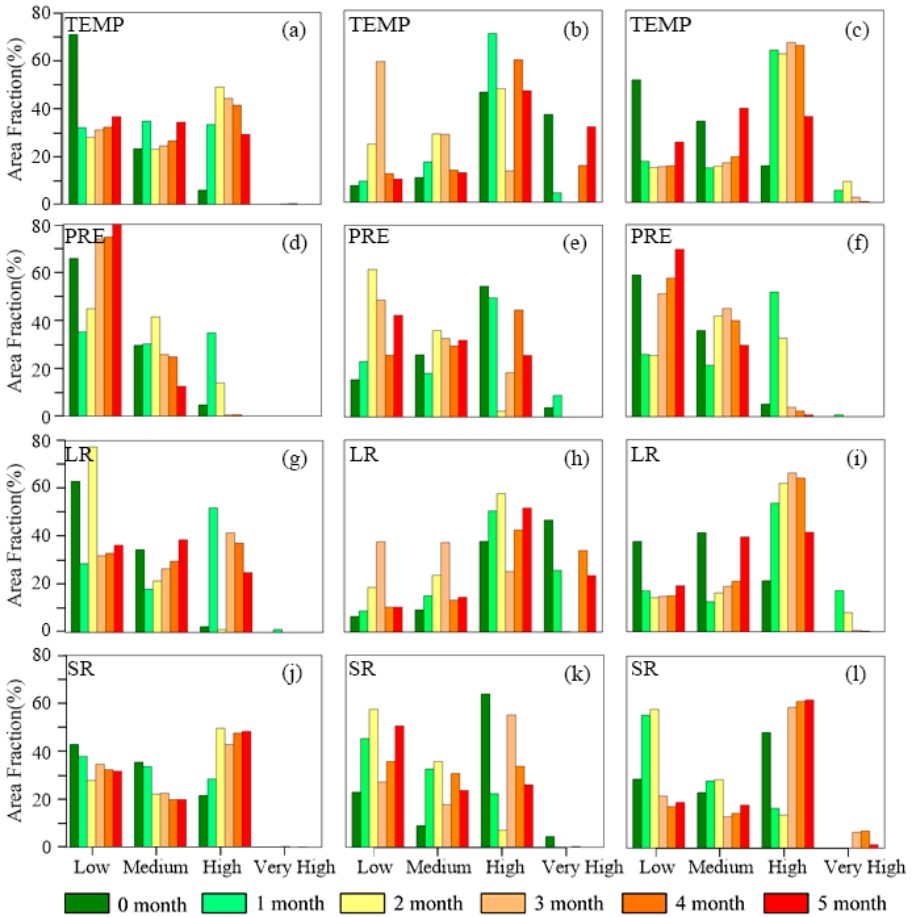

**Figure 7.** Statistics of area fractions of four levels under each climatic factor and each season: (**a**,**d**,**g**,**j**) for summer; (**b**,**e**,**h**,**k**) for autumn; (**c**,**f**,**i**,**l**) for the growing season.

### 3.3.2. Responses of Seasonal NDVI to Climatic Factors

Considering the time-lag effects of NDVI responses to different climatic factors in different seasons, the partial correlation coefficient between NDVI and each climatic factor was calculated.

In Summer, the positive partial correlations between precipitation, downward long-wave radiation, and NDVI were found in most of the HEM regions (Figure 8b–c). NDVI in Autumn negatively correlated with the temperature in the central region (Figure 9a), which was consistent with the study by Dong et al. [52]. This result is possibly because of the drought limitation. As shown in Figure 9c, there was a significant negative correlation between downward long-wave radiation and autumn NDVI in the southeastern HEM region, which is not consistent with that in other parts or other seasons (Summer and growing season). During the whole growing season (May–October), the NDVI exhibited positive correlations with temperature in the southern HEM region and negative correlations with temperature in Lhasa city and the surrounding areas.

Figure 10 depicts the spatial patterns of the main influencing factors, respectively, in Summer, Autumn, and the growing season. One of the four climatic factors with the largest absolute value of the partial correlation coefficient was identified as the main influencing factor (MICF) (Figure 11). In Summer, areas with precipitation and downward long-wave radiation as the main influencing factor account for 25.77% and 26.65% of the total areas, respectively, while those with temperature and downward short-wave radiation as the main influencing factors only account for 10.18% and 3.1% of the total areas, respectively. In Autumn, area proportion with downward long-wave radiation as the main influencing factor increases up to 58.18% compared with that in Summer. The areas with the main influencing factor being downward short-wave radiation and precipitation only account

for 14.57% and 6.99% of the total areas, respectively, and these were distributed in the western HEM region. Throughout the whole growing season (May to October), downward long-wave radiation plays the most critical important role in vegetation changes, followed by precipitation and downward short-wave radiation. The temperature has a minor influence on vegetation changes.

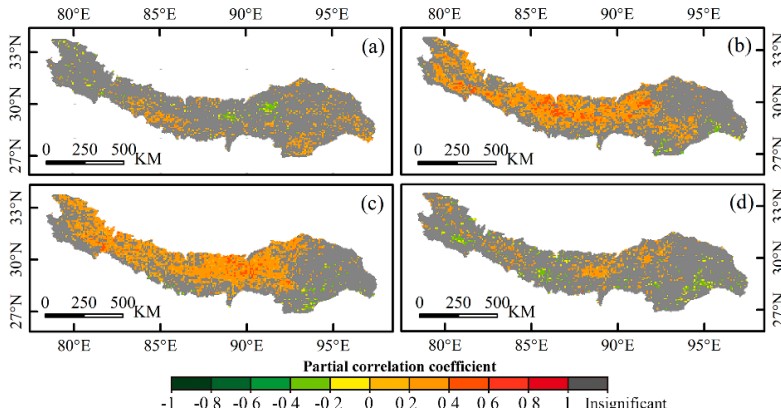

**Figure 8.** Spatial patterns of partial correlation coefficients between summer NDVI and four climatic factors: temperature (**a**), precipitation (**b**), downward long-wave radiation (**c**), and downward short-wave radiation (**d**).

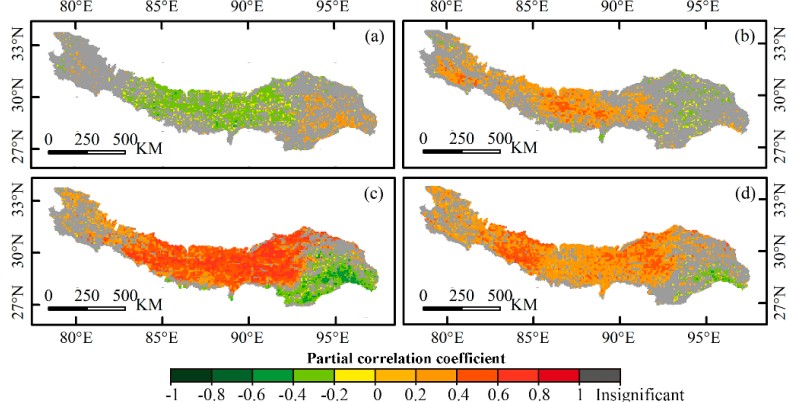

**Figure 9.** Spatial patterns of partial correlation coefficients between autumn NDVI and four climatic factors: temperature (**a**), precipitation (**b**), downward long-wave radiation (**c**), and downward short-wave radiation (**d**).

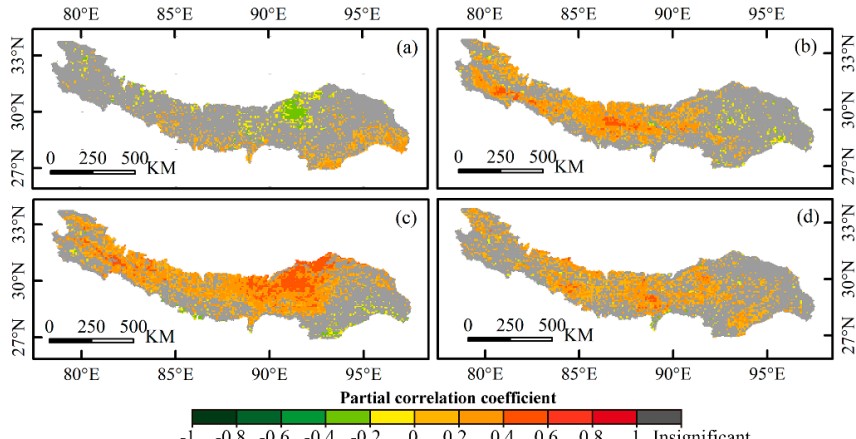

**Figure 10.** The spatial patterns of the main influencing factor on NDVI variations in summer, autumn, and the growing season.

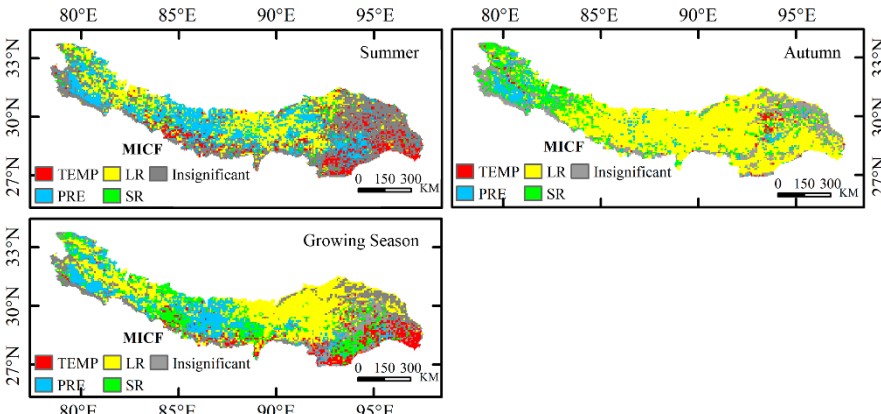

**Figure 11.** Spatial patterns of partial correlation coefficients between the growing season NDVI and four climatic factors: temperature (**a**), precipitation (**b**), downward long-wave radiation (**c**), and downward short-wave radiation (**d**).

## 4. Discussion

### 4.1. NDVI Trends and Breakpoint

Early studies showed that mountainous region in China exhibited greening trends [53], such as the increase in vegetation at a rate of 0.0024 year$^{-1}$ in the Changbai Mountain area [54], the positive NDVI trend at a rate of 0.0031 year$^{-1}$ in the the Qilian mountain area [55], and the improved vegetation at a rate of 0.003 year$^{-1}$ in the Altun Mountains [56]. The growth rate of NDVI in the HEM region is slower than that of other mountainous regions.

The breakpoint of NDVI in the HEM region was found to be around 1989, which is consistent with that in the North American [57,58] and the Asia-Pacific regions [58]. In the Asia-Pacific region, the detected breakpoint was 1991, based on the thirty-year time series of GIMMS NDVI, with an increasing rate of 26.14 × 10$^{-4}$ year$^{-1}$ before 1991 and 5.78 × 10$^{-4}$ year$^{-1}$ after 1991 [58]. Unlike the Asia-Pacific region, the interannual trend of NDVI in the HEM region was negative after 1989. This negative trend has been reported in other regions, such as temperate and boreal Eurasia [40,59], western North America [60], tropical regions [61], and China's Tibetan areas [59]. Temperature-induced moisture stress is possibly the main influential factor in the NDVI trend [59–62]. The precipitation and temperature trends were calculated before and after 1989. In the HEM region, the precipitation trend (0.3 mm·year$^{-1}$) before 1989 was higher than that after 1989 (0.16 mm·year$^{-1}$), and the temperature trend changed little. Temperature-induced moisture stress was also attributed to the negative trend in the HEM region.

NDVI trends at the pixel scale were heterogeneous. Most pixels with positive trends were concentrated in the Gongbujiangda county and the surrounding areas. These positive trends agree with the reported result that the largest increases in NDVI were found in the Niyang basin, which contains Gongbujiangda county [63]. Rising temperatures over the past decades might have possibly contributed to the positive trends of NDVI [16,64]. The local non-commercial forest management strategy has also contributed to these positive trends. The largest negative trends were found in the southeast portion of the HEM region. Zhang et al. [22] reported that broadleaf forests in southeastern Tibet plateau experienced the trend of degradation from 1982 to 2006. The decrease in sunshine duration resulted in a negative trend of NDVI in the southern plateau [65]. A similar phenomenon of vegetation degradation caused by declining solar radiation was found in other places, such as the Northern Tibet [66] and Amazon regions [67].

### 4.2. The Relationships between Interannual NDVI Trends and Environment Factors

In Section 3.2, two variables (temperature and shortest distance to large lakes) had the highest importance. The rising trend in temperature has been the main driving force for vegetation greening in the interannual time scale, possibly because the rising temperature accelerated the growth rate of vegetation in some areas, such as the Nyingchi region [68]. Meantime, temperature is considered the main influencing factor for vegetation in the Tibet plateau [69]. There is an inverted U-shaped relationship between the interannual NDVI trend and the shortest distance to lakes, that is, a positive correlation within twenty kilometers and negative correlation when exceeding twenty kilometers, which has also been found in other studies [70]. Li et al. [71] and Chen et al. [72] believed that there were two possible reasons for explaining the relationships. On the one hand, in summer, the lake effect causes a decrease in temperature in the vicinity of the lakes, which leads to an unsuitable temperature for the vegetation growth. On the other hand, when exceeding a certain distance, water vapor sourced from lake water evaporation decreased with an increase in distance from the lakes, which slows the vegetation growth rate.

Elevation, downward long-wave radiation (LR) and short-wave radiation (SR) have medium importance for interannual NDVI trends. As the elevation increased, the NDVI trend increased if the elevation was less than 5000 m, but it decreased if the elevation was higher than 5000 m. The above phenomenon could be ascribed to the noticeable elevation-dependent warming effect at elevations below 5000 m and the negligible elevation-dependent warming effect at elevations above 5000 m [73]. Yan and Liu [43] investigated the warming rates at different elevations in the Qinghai-Tibet Plateau and found that when the elevation is lower than 5000 m, the warming rate increases with elevation. The SR and LR are important parts of the ground energy balance. Under the increased SR trend and LR trend, the land surface gives up more moisture, making the ground drier [74]. This drier climate is not conducive to vegetation growth, so the NDVI trend is negatively correlated with the SR trend and LR trend. However, the drying effect is not apparent on a seasonal scale because the strengths of LR and SR are affected by seasonality [75].

### 4.3. Analysis of Time-Lag Effect for Different Climatic Factors

The time-lag effect reflects the sensitivity of vegetation to climates, which may vary with geographic locations and climatic factors. On a global scale, vegetation has a certain time-lag effect of more than one month with temperatures at low latitudes. At middle and high latitudes, vegetation generally responds quickly to temperature [18]. At the regional scale, the most mountainous vegetation is driven by temperature changes. In the Qilian Mountain areas, the time-lag effects of NDVI on temperature are shorter than one month in the growing season, Spring and Autumn, and longer than two months in Summer [55]. In the Central Himalayas of Nepal, there is no obvious time-lag effect between temperature and vegetation, possibly because the temperature is a limiting factor [76]. For the semi-arid grasslands in Tianshan mountain, the optimal time lag of NDVI to temperature is only twenty days, because much of the glacial meltwater induced by temperature meets vegetation demands [77]. In the HEM region, the time-lag effect of NDVI to temperature is longer than that in the Central Himalayas of Nepal and the Tianshan Mountain, which indicates that vegetation changes are not sensitive to temperature.

Precipitation is an important factor affecting vegetation changes. In southwestern American [78], the Australian outback [79], and the Yun-Gui plateau [80], vegetation changes are strongly positively correlated with precipitation preceding one month. However, in the Amazon Region [19] and the East Tibetan Plateau [24], the time lags are greater than one month. Regional climates play an essential role in the spatial difference of the time lags. Most of the HEM region belonged to arid and semi-arid climates, of which precipitation can effectively alleviate the strong demands of vegetation on water. Therefore, the vegetation of the HEM region is more sensitive to precipitation than to temperature.

Downward long-wave radiation and downward short-wave radiation are two key factors that influence the land surface ecosystem. Wang et al. [81] reported that in the permafrost region of the

Qinghai-Tibet plateau, vegetation growth is more sensitive to downward long-wave radiation than short-wave radiation. Pepin et al. [82] investigated the relationship between downward long-wave radiation and elevation-dependent warming effects on mountainous ecosystems and concluded that downward long-wave radiation contributes more to the warming effect at higher elevations or low humidity areas. Therefore, the high sensitivity of long-wave radiation to vegetation changes in the HEM region may be a result of the warming effect of long-wave radiation on vegetation at night.

*4.4. Partial Correlations between Seasonal NDVI and Climatic Factors*

In this study, it was found that the autumn NDVI has a negative correlation with temperature in the central HEM region. This result is consistent with the result reported by Zhang et al. [83], who found that a significant negative correlation between autumn NDVI and temperature was found in the southwest of the Tibet Plateau. Du et al. [16] found that summer NDVI was strongly negatively correlated with temperature, and autumn NDVI was weakly positively correlated with temperature, which is not consistent with these results. There are two reasons for explaining the above phenomenon. In the one hand, the summer vegetation responding to temperature has a time-lag effect—that is, vegetation will not change immediately with an increase in temperature. Simultaneously, much glacier meltwater and precipitation effectively supplements the lack of soil moisture and avoids drought restriction for vegetation [84,85]. On the other hand, in autumn, in the background of the decreased precipitation and decreased glacial meltwater, rising temperatures contribute to a drier climate, which hinders vegetation growth. Another interesting finding is that negative correlations between temperature and the growing season's NDVI were concentrated in Lhasa and the surrounding areas. With rapid urbanization, there has been degradation of natural vegetation in Lhasa and the surrounding areas [86]. Meanwhile, there is an anomalously high warming rate because of the urban island effect [87]. Therefore, local NDVI is negatively linked with temperature changes.

Downward long-wave radiation is vital for vegetation changes in Summer, Autumn, and the growing season. Walters et al. [88] reported that the changes in downward longwave radiation could mix the warm air and land surface to warm the surface temperature. High altitude areas are considered to be more prominent in this warming effect [82]. Rangwala et al. [75] found that downward long-wave radiation is related to winter warming effects in the Tibet plateau. Downward long-wave radiation is sensitive to changes in humidity. The higher the humidity, the greater the radiation, and vice versa [89]. Therefore, when the climate tends to be dry, the downward long-wave radiation decreases, which does not lead to much evaporation. The HEM region is characterized by a big day-night temperature difference in autumn. In this region, the increased downward long-wave radiation is beneficial to the warming effect at night, which protects vegetation from the frost.

**5. Conclusions**

Current related research has focused on the response of NDVI variation to climate change, while interannual NDVI trends and their influencing factors are rarely discussed. This paper analyzed the relationships between interannual NDVI trends and environmental factors, seasonal NDVI variations, and climatic factors in the HEM region. There was also a discussion on the time-lag effect of NDVI responses to climatic factors. The following conclusions were obtained:

(1) From 1982 to 2015, the overall NDVI of the HEM region exhibited a weak upward trend. In detail, the NDVI showed a significant and rapid upward trend before 1989 and a downward trend after 1989. At the pixel scale, many greening pixels were concentrated in Gongbujiangda county and the surrounding areas, because of the rising temperature, plenty of precipitation, and the local forest protection strategy.

(2) Among nine environmental factors, the interannual temperature trend and the closest distance to large lakes are the most important factors affecting the NDVI trends in the HEM region. The increasing temperature leads to an increase in the NDVI trend, possibly because rising temperature accelerates the rates of photosynthesis and respiration in vegetation. Within 20 km,

the shortest distance to large lakes is positively correlated with the NDVI trend. Glacial lakes in the HEM region show a cooling effect on the temperature of its nearby area, which may limit the vegetation growth to some extent. This correlation is negative when exceeding twenty kilometers because air humidity decreases with an increase in the distance to large lakes, which is not conducive to vegetation growth in the semi-arid region.

(3) In the HEM region, the time lags of NDVI responses to precipitation and downward long-wave radiation are short, and those to temperature and short-wave radiation are long. Seasonally, the time lags of NDVI to climate factors in autumn are shorter than that in summer.

(4) Autumn NDVI was negatively correlated with temperature in the central HEM region, possibly because of increasing temperature-induced moisture stress. A negative correlation between temperature and NDVI in the growing season was found in Lhasa and the surrounding areas, probably because of the urban heat island effect and intense human activities. Among four climatic factors, downward long-wave radiation was the main climate factor that influenced NDVI changes in Autumn and the growing season, possibly because of its warming effect at night.

**Author Contributions:** Z.-X.Y., Z.-Q.Z., and W.-M.C. conceived the manuscript; Z.-X.Y. wrote the manuscript; J.-Y.G., H.D., and N.W. reviewed and edited the manuscript.

**Funding:** This work was jointly supported by the Science and Technology Project of Xizang Autonomous Region (Grant No. XZ201901-GA-07), National Natural Science Foundation of China (Grant No. 41661144042 and 91647205) and the Fundamental Research Funds for the Central Universities of Chang'an University (No.300102278302).

**Conflicts of Interest:** The authors declare no conflict of interest.

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
