# Peer review of "Interannual and Seasonal Vegetation Changes and Influencing Factors in the Extra-High Mountainous Areas of Southern Tibet"

_remotesensing, doi:10.3390/rs11111392_

Round 1

Reviewer 1 Report

See attachment 

Reviewer 2 Report

I found the manuscript is interesting, well-conceived and well presented. However, I would like to recommend a few changes. The following points can be improved:

1.      As MVC function has been applied, the NDVI range should be close to 1; however the figure shows very narrow range of NDVI (0.27 to 0.30). It needs more description.   

2.      In Figure 2a, R2 = 0.03 but p <0.05, It may be not correct and authors should prove degree of freedom.    

3.      BFAST detected break point as 1989 and this is obvious for extreme jump. Why 1989/90 shows relatively large values. It could be due to narrow range of NDVI as indicated in point 1. Is there any references indicating 1989 as a breakpoint?  

WHY post 1989 ndvi trend declining despite forest deforestation rate was curbed. Authors should discuss possible causes.   

4.       In Fig 3, check the TEMP trend whether per year or per decade has been calculated?

5.      In Table 2, % variation explained for interannual NDVI trends of the whole study area was up to 40%, indicating some additional key parameters are not mentioned. The rationale on the choice of the particular set of parameters should be explained with more details.

6.      In Fig 5, both indicators, namely, Mean Decrease Gini (IncNodePurity) and Mean Decrease Accuracy (%IncMSE) showed similar values (y1 and y2 axis). The rationale on the choice of ncNodePurity and %IncMSE should be explained with more details.  The text between 252 to 256 may needs more clarity.

7.       Some inconsistencies and minor errors that needed attention. For instance L31: lag e spell check, etc.

Reviewer 3 Report

I have reviewed the manuscript remotesensing-493626 and I believe the authors have carried out an interesting study to determine the influence of different factors in the vegetation dynamics of southern Tibet.   

I consider that there are some details that should be improved in order to a better understanding of the manuscript. I have some suggestions to improve it. My comments are detailed below:

1.      I recommend the authors to revise the entire document in order to improve the writing.

a.       Sometimes, there are very long sentences.

b.      Revise the verbs in some sentences (i.g. L56-57; L81-82, L346-349…)

c.       Insert blank space (i.e. L106, L108, L143, L158, L159, L 168, L189…)

d.      Revise the need for articles (i.e. L 83 and Figure 1 “The study area…”; L106 “The spatial resolution…”)

e.       Try to insert the equations, the figures and the tables right after the paragraph they are commented.

f.       Equations (1) and (2) should be separated in the manuscript.

g.       Revise the references. Some of them are in different format than this journal remommend (i.e. L 162, L 217, L 311…)

h.      Revise the use of the word “Meanwhile” in the manuscript.

i.        Sometimes the sentences are too short and do not describe well what you want to show (i.e. L 245-248, L 335-336…)

j.        Revise the use of generic words like that (L436), one (L258)…

2.      L 27. Revise the way the units are showed. ºC/year should be ºC year-1. The same circumstance can be observed throughout the document (i.e. L200, L 469, Table 1…).

3.      In some lines, the authors wrote soil zonal instead soil zones (i.e. L30, L243, L243, L 476 and Table 2).

4.      L 31. Revise “time lag e”

5.      L81-99. I suggest the authors reorganize the information of both paragraphs. The information about the orography of the HEM should be the presentation card of the study area.

6.      L90. In the previous paragraph, authors indicated that Himalayas extra-high mountain will be called HEM hereinafter.  

7.      Why the authors change the spatial resolution from 0.0833º to 0.05º? This change in the spatial resolution is not advised because it is increasing the detail without real data. In remote sensing, it is recommended to make modifications to larger pixel sizes, but not to smaller.

8.      L 122-124. Some references are needed in this sentence.

9.      L 156. Authors should insert the full name of BFAST in this sentence (although it is explained in 2.2.3 section).

10.  L159. I suggest a full stop.

11.  L 175. Authors should insert the full name of %IncMSE and IncNodePurity in this sentence.

12.  Table 1. What is European distance? Is European distance the same as Euclidean distance?

13.  Table 1. The description of every factor does not follow a methodology and sometimes fails to describe the variable. (i.e. what classification is used for soil zone?)

14.  Table 1. Al the acronyms should be explained in a footer in order to read the table independently of the text.

15.  L200. Speed or Slope?

16.  L206-219; L323-327. Most of these paragraphs are discussion of the results. In the same way, L 379-385 can be considered as results. I suggest the authors to comment and discuss the results jointly.

17.  L 238-251. As mentioned above, the writing of the manuscript is sometime not very clear. Revise the paragraph.

18.  L 257-273. Try to reduce the number of times the authors say "trend" in the paragraph. It is very redundant.

19.  L 271-273. Are this data shown?

20.  L 296. Discussed? In the result section?

21.  L 303. Coefficient of determination (R2)

22.  L 336-338. This sentence should be in the next paragraph.

23.  L 356-358. Revise this sentence. It's not easy to understand.

24.  L410. Do not repeat the authors twice. I suggest something like that: “…with the findings of other studies [13, 24, 49]. Wu et al [24]…

25.  Revise the way Velázquez is written (L 69 and reference [22]).

Author Response

Responses to Reviewer 3   Comments

Point 1:

I recommend the authors to revise the entire document in order to improve the writing.

Response 1:

Thank you for your suggestion. Following your suggestions, we have revised the verbs, sentences, equations and references of the entire document. We have had our manuscript checked by a professional English editing service. 

Point 2:

L 27. Revise the way the units are showed. ºC/year should be ºC year-1. The same circumstance can be observed throughout the document (i.e. L200, L 469, Table 1…).

Response 2:

Thank you for your suggestion. We have revised this kind of error across the entire paper.

Revised 2:

Line 271,

temperature change, most areas showed a positive trend with a rate between 0 and 0.035℃·year-1.

Line 295

When the warming rate is higher than 0.013℃·year-1, the NDVI trend remains unchanged.

Table 1

of temperature (℃·year-1). Each pixel has an interannual trend.

Interannual precipitation trend (mm·year-1) was obtained by the

Interannual downward long-wave radiation trend (W·m-2·year-1)

Point 3: In some lines, the authors wrote soil zonal instead soil zones (i.e. L30, L243, L243, L 476 and Table 2).

Response 3:

Thank you for your suggestion. In section 3.2, we removed the content about the explanatory power of different classified variables, such as soil zones. The section 3.2 focused on the relationship between nine environmental variables and NDVI trends.

Point 4: L 31. Revise “time lag e”

Response 4:

Thank you for your suggestion. I have revised “time lag e”

Revised 4:

Line 31-32 The time-lag effects of NDVI response to four climatic factors are shorter in Autumn than in Summer.

Point 5: L81-99. I suggest the authors reorganize the information of both paragraphs. The information about the orography of the HEM should be the presentation card of the study area.

Response 5:

Thank you for your suggestion. I added the presentation card of study area in section 2.1.

Revised 5:

Line 95-109

The Himalayas extra-high mountain region (HEM) is located in the southern Tibet (Figure 1a). It stretches approximately 1700 kilometers from west to east and 1000 kilometers from south to north, with mean elevation more than 4000 meters.

From east to west, the HEM region has experienced a huge climate changes from a humid climate to a semi-humid climate, semi-arid climate and arid climate (Figure 1b). Medog county and Cona county are situated in the humid climate region with an annual precipitation above 500 mm [24], mainly covered with broad-leaved forests and needle-leaved forests. Gongbujiangda county is situated in the transition from southern Tibet valley to eastern Tibet alpine valley, with mild and humid climate in its eastern portion and cold and dry in its western portion. Coqen, Zhongba and Saga counties belonged to semi-arid regions with the annual precipitation ranging from 200 to 300 mm, characterized by large daily temperature fluctuation and long light duration. Alpine vegetation, grassland and meadow account for 80% of total areas in the semi-arid region. HEM is rich in water resources and contains six catchments, including upstream, midstream and downstream catchments of Yarlung Zangbo River, Zangnan inland river catchment, Zangxi inland river catchment and the Chang Tang grassland inland river catchment.

Point 6: L90. In the previous paragraph, authors indicated that Himalayas extra-high mountain will be called HEM hereinafter.

Response 6:

Thank you for your suggestion. I revised this kind of error across the paper.

Revised 6:

Line 98

From east to west, the HEM region has experienced a huge climate changes from a humid

Line 260

Figure 2a shows illustrates the interannual NDVI trend variation and its spatial distribution at the whole HEM scale from 1982 to 2015.

Line 394

were calculated for before and after 1989. In the HEM region, the precipitation trend (0.3 mm·year-1)

Line 404

found in the southeast portion of the HEM region. Zhang et al. [43] reported that broadleaf forest in

Point 7: Why the authors change the spatial resolution from 0.0833º to 0.05º? This change in the spatial resolution is not advised because it is increasing the detail without real data. In remote sensing, it is recommended to make modifications to larger pixel sizes, but not to smaller.

Response 7:

Thank you for your suggestion. we realize this problem. On the one hand, the purpose of this change is to match meteorological data. On the other hand, due to the introduction of catchment vectors, downward resampling is conducive to dividing all pixels into different catchments, and then input into Random Forests model

Point 8: L 122-124. Some references are needed in this sentence

Response 8:

Thank you for your suggestion. We added references to support related content in section 2.1.

Revised 8:

Line 134-140

GEWEX-SRB downward shortwave radiation, Princeton forcing data and GLDAS data. The CMFD has been widely utilized in plant primary productivity estimation [30], driving factor analysis of vegetation growth [31], and lake area simulation [32]. Before the utilization, the 3-hour interval meteorological dataset were aggregated into the monthly average temperature, monthly total precipitation, monthly total downward long-wave radiation and monthly total downward short-wave radiation data. Meanwhile, the processed data were resampled to 0.05° using nearest neighborhood method.

Point 9: L 156. Authors should insert the full name of BFAST in this sentence (although it is explained in 2.2.3 section).

Response 9:

Thank you for your suggestion. We have revised this error for a clear expression.

Revised 9:

Line 173-175

Compared with the commonly used method for time series analysis, such as Fourier spectral analysis and Breaks for Additive Season and Trend (BFAST)

Point 10:   L159. I suggest a full stop.

Response 10:

Thank you for your suggestion. We have revised the content in section 2.3.1. We put the specific operation process at the beginning of section 2.3.1.

Revised 10:

Line 151-159

EEMD algorithm were used to extract the annual components of NDVI and climatic factors at the whole HEM scale and pixel scale. Taking NDVI as an example, the monthly average NDVI values from 1982 to 2015 were calculated to generate a time series X(t) with 408 values. Then, the time series of X(t) was input into the EEMD algorithm to generate m IMF components and one residual (Eq. (1)). Each IMF has its own mean period T, which can be calculated by Eq. (1). Based on the grouping criteria proposed by Wen et al. [33], we summed one residual and the IMFs with the mean period T greater than 2 to obtain the interannual variation component. if needed to obtain the interannual trend, linear regression was applied for the annual variation component. At pixel scale, the same method was used to extract the interannual variation component pixel-by-pixel.

Point 11:  L 175. Authors should insert the full name of %IncMSE and IncNodePurity in this sentence.

Response 11:

Thank you for your suggestion. we added the full name of %IncMSE in Line 216. We finally selected the %IncMSE to evaluate the importance of variables. The principle of %IncMSE is closer to random testing and %IncMSE can reflect the importance of each variable in all variables.

Revised 10:

Line 218-221

The main evaluation indice for the importance are mean decrease accuracy (%IncMSE). %IncMSE means the percent increase in MSE as a result of the variable being randomly permuted. More important factor has the higher values of %IncMSE.

Point 12: Table 1. What is European distance? Is European distance the same as Euclidean distance? 

Response 11:

Thank you for your suggestion. We used the variables are the Euclidean distance to large lakes and the Euclidean distance to rivers

Revised 10:

In Table 1

Euclidean distance (m) to the nearest rivers

Euclidean distance (m) to the nearest lakes

Point 13: Table 1. The description of every factor does not follow a methodology and sometimes fails to describe the variable. (i.e. what classification is used for soil zone?)

Response 13:

Thank you for your suggestion. We revised Table 1. For interannual temperature trend, interannual precipitation trend, interannual long-wave radiation trend and interannual short-wave radiation trend, we added the description about the calculation process. For the distance to rivers and the distance to lakes, we added the descriptions of lakes and rivers. We removed the soil zone and vegetation type because they do not occur in the late discussion.

Revised 13:

Table 1

Table 1. Descriptions of nine environmental factors

Factor

Description

Interannual   temperature trend

Firstly,   interannual variation component of temperature was extracted by the EEMD   algorithm. Subsequently, the linear regression was applied for the   interannual variation component to obtain the interannual trend of temperature (℃·year-1).   Each pixel has one interannual trend.

Interannual   precipitation trend

Interannual precipitation trend (mm·year-1)   was obtained by the same method   as above.

Interannual   downward long-wave radiation trend

Interannual downward long-wave radiation trend (W·m-2·year-1)   was obtained   by the same method as above.

Interannual   downward short-wave radiation trend

Interannual   downward short-wave radiation trend (W·m-2·year-1)   was obtained by the same method as above.

Elevation

The digital   elevation model (DEM) with a spatial resolution of 0.05°

Slope

Slope (°) was   calculated from DEM through the surface analysis function of the ArcGIS   software

Distance to rivers

Euclidean   distance (m) to the nearest rivers >100 m

Distance to   large lakes

Euclidean distance (m) to the nearest lakes > 1000 m2

Catchment

China   third-level catchment boundary was used in the study. The HEM region was   divided into seven sub-regions, namely the Chang Tang Grassland Inland River   catchment, Upstream catchment of Yarlung Zangbo River, Midstream   catchment of Yarlung Zangbo River, Downstream catchment of Yarlung Zangbo   River, Zangxi Inland River catchment and Zangnan Inland River catchment.

EEMD: Ensemble Empirical Mode Decomposition

Point 14: Table 1. Al the acronyms should be explained in a footer in order to read the table independently of the text

Response 14:

Thank you for your suggestion. We have revised this error. An annotation was added at the footer of Table 1.

Revised 14:

Line 228

EEMD: Ensemble Empirical Mode Decomposition

Point 15: L200. Speed or Slope?

Response 15:

Thank you for your suggestion. It refers to speed. We have revised inconsistent sentences about speed and slope .

Revised 15:

Line 261-265

The overall NDVI exhibited a statistically non-significant positive trend between 1982 and 2015 at a speed of 0.00012 year-1 (p>0.05), slower than the growth of Tibet Plateau (0.0002 year-1) [52]. The breakpoint in interannual NDVI variations appeared in 1989 (Figure 2b). Before the breakpoint time, NDVI exhibited a consistent positive trend with a rate of 0.0015 year-1 (p<0.01). After the breakpoint time, a negative trend with a speed of 0.00044 year-1 (p <0.01) was found.

Point 16: 16.  L206-219; L323-327. Most of these paragraphs are discussion of the results. In the same way, L 379-385 can be considered as results. I suggest the authors to comment and discuss the results jointly

Response 16:

Thank you for your suggestion. In the result and discussion section, we made a lot of modifications to separate the result from the discussion. The discussion section was strengthened.

Revised 16:

Line 270-274 (Result section)

At the pixel scale, significant positive trends of interannual NDVI component were found in Gongbujiangda county and surrounding areas, and weak positive trends with a speed between 0 and 0.0005 year-1 were found in central and northwestern HEM region (Figure 3). Pixels with a negative trend of interannual NDVI were concentrated in the southeast HEM region.

Line 398-408 (Discussion section)

NDVI trends at pixel scale have been heterogeneous. Most pixels with positive trends were concentrated in the Gongbujiangda county and surrounding areas. These positive trends agree with another study that reported that the largest increases of NDVI were found in the Niyang basin which contained Gongbujiangda county [64]. Rising temperatures over the past decades might have possibly contributed to the positive trends of NDVI [19,65]. Local non-commercial forest management strategy has also contributed to these positive trends. The largest negative trends were found in the southeast portion of the HEM region. Zhang et al. [43] reported that broadleaf forest in southeastern Tibet plateau experienced the trend of degradation from 1982 to 2006. The decrease in sunshine duration was supposed to result in a negative trend of NDVI in the southern plateau [66]. The similar phenomena of vegetation degradation caused by the declined solar radiation are found in other places, such as Northern Tibet [67] and Amazon region [68]

Point 17: L 238-251. As mentioned above, the writing of the manuscript is sometime not very clear. Revise the paragraph.

Response 17:

Thank you for your suggestion. We remove the content about explanatory power and categorical variables (L 238-251), and the emphasis is put on describing the relationship between environmental variables and NDVI trends.

Point 18: L 257-273. Try to reduce the number of times the authors say "trend" in the paragraph. It is very redundant.

Response 18:

Thank you for your suggestion. We re-phrased the sentences to reduce the use of this word .

Revise

Line 287-308

Point 19: L 271-273. Are this data shown?

Response 19:

Thank you for your suggestion. The data is shown in Figure 6. At same time, we add the description of catchments in the “Study Area” section and Figure 1b.

Revise

Line 106-109

The HEM region is rich in water resources and contains six catchments, including upstream, midstream and downstream catchments of Yarlung Zangbo River, Zangnan inland river catchment, Zangxi inland river catchment and the Chang Tang grassland inland river catchment.

Point 20: L 296. Discussed? In the result section?

Response 20:

Thank you for your suggestion. We revised the discussion section to explain what the characteristics of the HEM region are and compared it with other places.

Revise 20:

Line 381-495

Point 21: L 303. Coefficient of determination (R2)

Response 21:

Thank you for your suggestion. We revised it in line 242-243.

Revise 21

Line 245

These correlation coefficients (R2) are divided into four levels according to the magnitude of their absolute values

Point 22: L 336-338. This sentence should be in the next paragraph.

Response 22:

Thank you for your suggestion. We move this content to Methodology section, which facilitate a more detailed description of the calculation process.

Revise 22

Line 243-247

Summer was taken as an example. When the time lag i=0,1,…,5 month, the correlation coefficients (R2) between NDVI and each climatic factor were calculated respectively. These correlation coefficients (R2) are divided into four levels according to the magnitude of their absolute values: Low (< 0.3), Medium (0.3-0.5), High (0.5-0.8) and Very-High (> 0.8). For each climatic factor and each time lag, we calculated the area fractions corresponding to four levels respectively.

Point 23:  L 356-358. Revise this sentence. It's not easy to understand.

Response 23:

Thank you for your suggestion. We rephrased the sentences between line 356 and line 358.

Revise 23

Line 351-354

During the whole growing season (May to October), Downward long-wave radiation plays the most important role in vegetation changes, followed by precipitation and downward short-wave radiation. Temperature have minor influence on vegetation changes.

Point 24: L410. Do not repeat the authors twice. I suggest something like that: “…with the findings of other studies [13, 24, 49]. Wu et al [24]…

Response 24:

Thank you for your suggestion. We reorganized the content about time-lag effect of NDVI on precipitation.

Revise 24

Line 449-456

Precipitation is one of the important factors affecting vegetation changes. In southwestern American [79], the Australian outback [80], and the Yun-Gui plateau [81], vegetation changes are strongly positively correlated with precipitation preceding one month. However, in the Amazon Region [22] and the East Tibetan Plateau [24], the time lags are greater than one month. Regional climates play an important role in the spatial difference of the time lags. Most of the HEM region belonged to arid and semi-arid climates, of which precipitation can effectively alleviate the strong demands of vegetation on water. Therefore, vegetation of the HEM region is more sensitive to precipitation than to temperature.

Point 25: Revise the way Velázquez is written (L 69 and reference [22]).

Response 25:

Thank you for your suggestion. We revised the problem and re-examined the format of the references.

Round 2

Reviewer 1 Report

The authors have made critical improvements to this paper and addressed all my concerns. I believe it is now ready for publication. Some minor details are described below:

·      I suggest reducing the last sentence of the abstract to: “This study is of great significance to revealing the relationship between vegetation and environmental factors in the mountainous areas.” The title already mentions southern Tibet, and the first sentence of the abstract mentions the Himalayas. Leaving the concluding sentence of the abstract broader could help put this research in a broader context. While this research is of great importance locally, I believe scientists working on other mountainous regions of the world, such as the Andes or African mountains, would find this study valuable and possibly replicable in their areas. They could also mention at the end of the introduction that this research “improves our understanding of vegetation change in mountainous regions” if desired.  

·      Check out spelling mistakes. For instance, 2.2 heading should be “2.2 Data source”, not “2.2. Date source”.

·      Show consistency in the use of terms. In sections 2.2.1 to 2.2.3 use either “data” or “dataset” in all of them.

·      Check out punctuation and capitalization. For instance, in line 157 there is an “if” (no caps) after a period.

·      Check consistency of tenses. In some sections, such as 2.3.2 and 2.3.3, past tense is used to describe the methods. Present tense is used in 2.3.4.

·       I suggest phrasing section 2.3.5 like the other methods section (i.e. 2.2.1 to 2.3.4), mentioning the use of the technique in the research and then detailing specific technical aspects of it. In section 2.3.5 a definition of correlation is provided first and then it’s used in the research is detailed.

Author Response

Responses to Reviewer 1 Comments

Point 1:

I suggest reducing the last sentence of the abstract to: “This study is of great significance to revealing the relationship between vegetation and environmental factors in the mountainous areas.” The title already mentions southern Tibet, and the first sentence of the abstract mentions the Himalayas. Leaving the concluding sentence of the abstract broader could help put this research in a broader context. While this research is of great importance locally, I believe scientists working on other mountainous regions of the world, such as the Andes or African mountains, would find this study valuable and possibly replicable in their areas. They could also mention at the end of the introduction that this research “improves our understanding of vegetation change in mountainous regions” if desired.

Response1: Thank you for your suggestion. We revised the concluding sentence of the abstract broader, which aims to improve the significance of this paper.

Revise 1:

Line 33-34

This study is of great significance to improves the understanding of vegetation change in mountainous regions.

Point 2:

Check out spelling mistakes. For instance, 2.2 heading should be “2.2 Data source”, not “2.2. Date source”.

Response 2: Thank you for your suggestion. We revised this error.

Revise 1:

Line 110

2.2 Data source

Point 3:

Show consistency in the use of terms. In sections 2.2.1 to 2.2.3 use either “data” or “dataset” in all of them.

Response 3: Thank you for your suggestion. We revised these terms in section 2.2.1 to 2.2.3.

Revise 1:

Line 111

2.2.1 GIMMS NDVI data

Line 126

2.2.2 Meteorological Data

Line 127

For this study, we utilized the China Meteorological Forcing Data (CMFD)

Line 131

data combines five auxiliary data sources: China Meteorological station data

Point 4:

Check out punctuation and capitalization. For instance, in line 157 there is an “if” (no caps) after a period.

Response 4: Thank you for your suggestion. We try to revise this kind of errors  

Revise 4:

Line 26:

(2) interannual temperature trends

Line 32:

(4) downward long-wave radiation was the main climate factor that influenced NDVI changes

Line 155:

If needed to obtain the interannual

Line 92

2.1.

Line 110

2.2. Data source

Line 111

2.2.1.

Line 126

2.2.2.

Line 139

2.2.3.

Line 182

Then the parameter of h was set to 1/7

Line 283-284

Spatial patterns of interannual trends of four climate factors: (a) temperature, (b) precipitation, (c) downward long-wave radiation, (d) downward short-wave radiation.

Line 336

NDVI responses to SR. The optimal time lags in summer, autumn and the growing season are two

Line 378

the growing season.

Line 494-495

and it protects vegetation from the frost.

Line 498

while interannual NDVI trends and their influencing factors are rarely discussed.

Point 5:

Check consistency of tenses. In some sections, such as 2.3.2 and 2.3.3, past tense is used to describe the methods. Present tense is used in 2.3.4.

Response 5: Thank you for your suggestion. We checked the consistency of tenses.

Revise 5:

Line 229

Simple linear regression was established to determine the correlation

Line 231

The relationships between NDVI and climatic factors were shown in

Line 243

These correlation coefficients (R2) were divided into four levels

Point 6:

I suggest phrasing section 2.3.5 like the other methods section (i.e. 2.2.1 to 2.3.4), mentioning the use of the technique in the research and then detailing specific technical aspects of it. In section 2.3.5 a definition of correlation is provided first and then it’s used in the research is detailed.

Response 6: Thank you for your suggestion. We added the content about the use of the technique in section 2.3.5.

Revise 6:

Line 247-252

Partial correlation is defined as the correlation of two factors controlling the influence of the other factors [50]. At the pixel scale, monthly NDVI and monthly climatic factor (temperature, precipitation, LR, SR) were input into the partial correlation analysis model in MATLAB, and then the partial correlation coefficients between NDVI and each climate factor were output. Subsequently, the climatic factor with the largest partial coefficient was selected as the main influence factor. The specific formula about the partial correlation was descripted as follows:

Reviewer 3 Report

I have reviewed the resubmitted manuscript remotesensing-493626 (revision 2) and the authors have clarified all the questions and comments suggested in the previous revision. Thanks for the efforts to improve the manuscript.

I only have a minor comment detailed below:

       Minor comment:

Table 1. As I previously commented, the description of the factors are no real explanation of variables, i.e. the information the variables offer. Authors are more interesting in including the description about the calculation process than the description of the own variable. In this case, I suggest changing the name of the field of the table to one more related to the calculation process.

Author Response

Point 1:

Table 1. As I previously commented, the descriptions of the factors are no real explanation of variables, i.e. the information the variables offer. Authors are more interesting in including the description about the calculation process than the description of the own variable. In this case, I suggest changing the name of the field of the table to one more related to the calculation process.

Response1:Thank you for your suggestion. We revised the head name of Table 1.

Revised 1:

Line 225

Table 1. Preprocessing of the nine environmental factors for the Random Forest model
